# Dermatopathology of Malignant Melanoma in the Era of Artificial Intelligence: A Single Institutional Experience

**DOI:** 10.3390/diagnostics12081972

**Published:** 2022-08-15

**Authors:** Gerardo Cazzato, Alessandro Massaro, Anna Colagrande, Teresa Lettini, Sebastiano Cicco, Paola Parente, Eleonora Nacchiero, Lucia Lospalluti, Eliano Cascardi, Giuseppe Giudice, Giuseppe Ingravallo, Leonardo Resta, Eugenio Maiorano, Angelo Vacca

**Affiliations:** 1Section of Molecular Pathology, Department of Emergency and Organ Transplantation (DETO), University of Bari “Aldo Moro”, 70124 Bari, Italy; 2LUM Enterprise srl, S.S. 100-Km.18, Parco il Baricentro, 70010 Bari, Italy; 3LUM—Libera Università Mediterranea “Giuseppe Degennaro”, S.S. 100-Km.18, Parco il Baricentro, 70010 Bari, Italy; 4Centro Interdisciplinare Ricerca Telemedicina—CITEL-, Università degli Studi di Bari “Aldo Moro”, 70124 Bari, Italy; 5Unit of Pathology, Fondazione IRCCS Ospedale Casa Sollievo della Sofferenza, 71013 San Giovanni Rotondo, Italy; 6Section of Plastic Surgery, Department of Emergency and Organ Transplantation (DETO), University of Bari “Aldo Moro”, 70124 Bari, Italy; 7Section of Dermatology and Veneorology, Department of Biomedical Sciences and Human Oncology (DIMO), University of Bari “Aldo Moro”, 70124 Bari, Italy; 8Department of Medical Sciences, University of Turin, 10094 Turin, Italy; 9Pathology Unit, FPO-IRCSS Candiolo Cancer Institute, Str. Provinciale 142 km 3.95, 10060 Candiolo, Italy

**Keywords:** artificial intelligence, AI, malignant melanoma, skin, software, algorithms, fast random forest (FRF)

## Abstract

The application of artificial intelligence (AI) algorithms in medicine could support diagnostic and prognostic analyses and decision making. In the field of dermatopathology, there have been various papers that have trained algorithms for the recognition of different types of skin lesions, such as basal cell carcinoma (BCC), seborrheic keratosis (SK) and dermal nevus. Furthermore, the difficulty in diagnosing particular melanocytic lesions, such as Spitz nevi and melanoma, considering the grade of interobserver variability among dermatopathologists, has led to an objective difficulty in training machine learning (ML) algorithms to a totally reliable, reportable and repeatable level. In this work we tried to train a fast random forest (FRF) algorithm, typically used for the classification of clusters of pixels in images, to highlight anomalous areas classified as melanoma “defects” following the Allen–Spitz criteria. The adopted image vision diagnostic protocol was structured in the following steps: image acquisition by selecting the best zoom level of the microscope; preliminary selection of an image with a good resolution; preliminary identification of macro-areas of defect in each preselected image; identification of a class of a defect in the selected macro-area; training of the supervised machine learning FRF algorithm by selecting the micro-defect in the macro-area; execution of the FRF algorithm to find an image vision performance indicator; and analysis of the output images by enhancing lesion defects. The precision achieved by the FRF algorithm proved to be appropriate with a discordance of 17% with respect to the dermatopathologist, allowing this type of supervised algorithm to be nominated as a help to the dermatopathologist in the challenging diagnosis of malignant melanoma.

## 1. Introduction

After the foundations of artificial intelligence (AI) were established, as postulated by the mathematician Alan Mathison Turing [1], the study and the analysis of AI underwent an important halt under the weight of expectations until the end of the last century; from these years, indeed, a new wave of technological development allowed the functional enhancement of computers, which became more and more powerful and complex [2]. Even today it is not clear whether a machine (computer) could be able to think, but certainly the development of increasingly complex systems, such as convolutional neural networks (CNNs) [3], has allowed a rapid increase in performance not only in fields of computer science, but also in different fields, such as medical, pathological and even subclassified dermatopathological fields [4,5].

The first attempts based on convolutional neural networks (CNNs) have already proven to be up to expectations, but the objective difficulty in reproducibility of the criteria for labeling a lesion as malignant or benign has also been reflected in the training of AI applied to differential diagnostics of atypical pigmented lesions. [3,4]. Indeed, it is not always easy to diagnose melanoma by analyzing the images with a “naked eye” approach. The problem has been known for some time and is classically defined as the “gray zone” of dermatopathology: in fact, beyond the extremes of the spectrum of benign and malignant, there is a fairly large number of atypical pigmented lesions that it is not easy to define “tout-court“ as having benign or malignant biological behavior. This uncertainty is the consequence of the fact that the criteria normally used to define a lesion as malignant melanoma become conflicting and blurred in certain situations; for example, nevi with severe dysplasia, Spitz nevi with dysplastic aspects (so-called SPARK nevus) or melanocytic lesions with an uncertain potential for malignancy such as MELTUMP or SAMPUS. The loss of criteria defined and recognized by all, as well as the interpretative subjectivity, has repercussions on the reduction or, sometimes, loss of agreement between dermatopathologists, and also (as some studies have highlighted) even on intraobserver agreement over time [2,3,4].

In this work we tried to train the proposed fast random forest (FRF) algorithm to be able to support the specialist to highlight automatically the “anomalous pixel regions” and to estimate a possible risk by quantifying the percentage of these regions with atypical morphological features starting from routine histopathological images (digital pathology).

## 2. Materials and Methods

### 2.1. Data Acquisition

For training, validation and testing, we used a dataset of 125 photomicrographs of 63 patients suffering from malignant melanoma, originally taken at 1280 × 1080 pixels (SONY^®^ Sensor IMX185, Tokyo, Japan) at 10× magnification, obtained from blocks of material fixed in formaldehyde and embedded in paraffin (FFPE) from 1 January 2020 to 1 January 2021. For each patient, demographic and clinical characteristics, including the Breslow thickness and other histopathological features, were recorded for routine clinical practice. Informed consent was obtained from all patients involved and the study was approved by the local ethical committee. In total, 63 specimens (mean length ± SD 1.64 ± 0.93 cm) were processed. For all specimens, hematoxylin and eosin (H&E) staining was used for histopathological analysis and, in some cases, ancillary immunohistochemical techniques were performed for a final correct diagnosis. For each of them, each individual case was analyzed and a more representative slide was chosen by two dermatopathologists with a lot of experience in skin histopathology (G.C. and A.C.). After collecting all the slides, we proceeded to identify at least 5 potential defects (hotspots) normally used in routine dermatopathological diagnostics to differentiate a dysplastic nevus from a malignant melanoma. The defects analyzed were divided into architectural (disposition of the single and aggregates of melanocytes, symmetry/asymmetry of the lesion in question) and cytological (nuclear atypia, pagetoid spreading of the melanocyte, possible necrosis) groups following the Spitz–Allen criteria [6] (Table 1).

### 2.2. Fast Random Forest (FRF) Image Classification

The artificial intelligence image processing algorithm used to classify and to enhance anomalies contained in the microscope image is the fast random forest (FRF) algorithm. The algorithm was designed by using a Java-based script framework. The learning process of the algorithm is based on a preliminary classification of clusters of pixels of the same image [5] including possible melanoma areas: the preliminary identification of melanoma morphological features represents the labeling approach typical of machine learning-supervised algorithms. The FRF testing provides as output the processed image with color-enhanced melanoma pixel clusters (each class selected in the learning step is represented by a color), probabilistic maps (high probability highlighted with white to identify an anomaly in a specified image region) and algorithm performance indicators (precision, recall, and receiver operating characteristic (ROC) curves [5]). The FRF algorithm executes an ensemble of decision trees able to classify clusters of pixels constituting an image with good accuracy, low computational cost and performing a multiclass segmentation [7,8,9,10,11,12]. The classification process is addressed by labeling clusters of pixels defined by the same features (clusters of gray pixel intensities). At the beginning, decision trees randomly select clusters of pixels by splitting the features at each node. The final classification of image areas are a result of the average classification of all the trees constructed during the algorithm iterations. The method is sketched in Figure 1, where it is possible to distinguish the following phases:Different clusters of pixels (features) are used for the training model;A marked region is distinguished as the features to find in the same image to process;The RF is executed by finding similar features in the same image (similar features of similar clusters having a similar gray pixel intensity distribution).

The training model can be constructed by setting the following tools or procedures (algorithm parameters):
Gaussian blur filtering, obtaining a blurred image to process;Sobel filtering, which is able to approximate the image by a gradient of the intensity;Hessian filtering, defined as:
(1)$$H=\left(\begin{array}{cc} h_{1} & h_{2}\\ h_{3} & h_{4} \end{array}\right)$$
where *h*1, *h*2, *h*3 and *h*4 are the elements of the Hessian matrix formed by the second-order partial derivatives stating the variation of the intensity of each pixel (derivative at the pixel point) in the x, y and xy direction (plane directions of the 2D image).

Defined by a *Trace*, a determinant (*Det*), a first eigenvalue (*Fe*), a second eigenvalue (*Se*), an orientation (*Or*; angle of the maximal second derivative), a *Gamma*, and a *Square Gamma*:(2)$$Trace=h_{1}+h_{4}$$
(3)$$Det=\sqrt{h_{1}{}^{2}+h_{2}h_{3}+h_{4}{}^{2}}$$
(4)$$Fe=\frac{h_{1}+h_{4}}{2}+\sqrt{\frac{4h_{2}{}^{2}+{\left(h_{1}-h_{4}\right)}^{2}}{2}}$$
(5)$$Se=\frac{h_{1}+h_{4}}{2}-\sqrt{\frac{4h_{2}{}^{2}+{\left(h_{1}-h_{4}\right)}^{2}}{2}}$$
(6)$$Or=\frac{1}{2}\arccos(4h_{2}{}^{2}+{\left(h_{1}-h_{4}\right)}^{2})$$
(7)$$Gamma=t^{4}{\left(h_{1}-h_{4}\right)}^{2}({\left(h_{1}-h_{4}\right)}^{2}+4h_{2}{}^{2})\;\mathrm{with}\;t=1^{3/4}$$
(8)$$Square\;Gamma=t^{2}({\left(h_{1}-h_{4}\right)}^{2}+4h_{2}{}^{2})\;\mathrm{with}\;t=1^{3/4}$$
Difference of Gaussian functions;Membrane projections due to the rotation of the original image kernel;Main pixel parameters (mean, minimum, maximum, etc.);Anisotropic diffusion filtering preserving sharp edges;Bilateral filtering preserving edges;Lipschitz filtering (preserving edges and decreasing noise);Gabor filtering (mainly adopted for texture analysis);Derivative filtering estimating high order derivatives;Structured filtering estimating the eigenvalues;Shifting of the image in different directions.

The optimized hyperparameters and filter properties applied for the image FRF processing (feature training) are: the Gaussian blur filter, Hessian matrix filter, membrane projections, membrane thickness equal to 1, membrane patch size equal to 19, minimum sigma equal to 1, and maximum sigma equal to 16.

In the proposed approach, the same image is adopted both for the training and for the testing: each image is converted into a grayscale image, and in the same image [12] the classes training the model are identified (classes of similar features including those of the Spitz nevi).

Different algorithms can be applied to calculate automatically the different areas [13]. In order to enhance classified clusters in the probabilistic image, a threshold filtering adjusting the intensities of the output grayscale image was adopted (see example of Appendix A): this setting is important to estimate with good precision the areas enclosing dermatological defects. The testing images were preliminary selected, taking into account images with certain defects in order to train the FRF algorithm efficiently.

The adopted method is summarized by the following steps:-A preliminary selection of images focusing the attention on the characteristics of dysplastic nevi and malignant melanoma (validation of the training model focusing on this classification);-Performing of a training of the selected images (FRF training model based on the classification of anomalous image areas embedding features of dysplastic nevi and MM, and the identification of other no-risk areas as structures of clusters of grayscale pixels);-Setting the optimization of the FRF algorithm parameters for the best identification of classes of the testing dataset;-Testing the execution of the FRF algorithm’s detection and estimation of anomalous areas by applying, after the analysis, image threshold filters (for the calculus, all the images have the same dimension of 1000 pixels × 2000 pixels);-Verification of the algorithm performance by estimating its precision.

## 3. Results

For five pixel clusters of the same dimensions (closest to the particular anomaly), a number of about 300 instances (computational cycles) occurs to achieve the maximum precision (equal to 1), with a computational cost of about 2 min using a processor, the Intel(R) Core(TM) i5-7200U CPU, 2.71 GHz. The minimum recall performance parameter (near to 0) is achieved in about 392 instances. The ROC curve (representing the true positive rate versus the false positive rate in the plane) is matched with the ideal curve of a perfect classifier (Figure 2). The performance indicators confirm the correct setting of the FRF hyperparameters.

Appendix B illustrates the FRF probabilistic images of some of the classified areas. The algorithm performance is estimated by the precision parameter representing the metric of the FRF score (probability of algorithm error). A maximum precision of 1 is achieved after about 250 iteration steps (see Figure 2).

An example of image classification is illustrated in Figure 3 where C1, C3 and C4 are the four classes used to construct the learning model (areas having similar characteristics).

In Table 2 and Table 3 the FRF enhancement of possible classified defects are shown by estimating the percentage coverage in 1 mm × 2 mm images.

Analyzing the discrepancy between the performance of the FRF algorithm and the diagnosis by the dermatopathologist, a value of 17% was found, which is slightly lower than that described in the interobserver variability (about 25–26%).

## 4. Discussion

Historically, the histopathological diagnosis of malignant melanoma has always fluctuated from rather simple and easy to classify cases up to very complex and difficult to interpret cases [14], considering that, in the context of human pathology, MM is defined as the “great mime” [15]. For example, in a recent 2017 paper, *Elmore G.J.* et al. [15] analyzed the diagnostic results of 187 practicing pathologists in 10 states by comparing them with each other and with a consensus diagnosis reached by a group of three experienced dermatopathologists. The authors asked the 187 pathologists to interpret the same skin lesions after a certain time range (8 months) in order to have an estimate of the intraobserver readability. The degree of accuracy was very high (about 92%) in the case of diagnosing slightly atypical pigmented lesions and reached about 72% in the case of invasive melanomas. Conversely, diagnostic accuracy became much lower in the case of lesions in the spectrum between these two extremes. For example, fewer than half of diagnoses agreed with expert consensus for cases classified as severely atypical lesions, melanoma in situ, or early stage invasive melanoma. Similarly, over time, the diagnoses of the so-called “gray zone” of dermatopathology lost intraobserver reproducibility [16].

In this context, the development of artificial intelligence methods applied to pathological anatomy [17] had the merit of offering a possibility to understand if the evaluation of a melanocytic lesion that represented a malignant melanoma could be made more “objective” compared to lesions not endowed with the potential for malignancy “tout-court”.

This work is based on the definition of the steps to follow to classify malignant melanoma defects. A metric estimating the precision of the applied FRF algorithm is applied to estimate the error of the classification. The low computational cost related the image processing and the use of the same image for the algorithm training allow us to apply the FRF algorithm to new images without constructing a training model based on historical images. In this way, we used a set of images of lesions previously diagnosed by two board-certified dermatopathologists to understand how much the FRF algorithm was able to assist the pathologist in the decision-making process.

The precision achieved by the FRF algorithm proved to be appropriate, allowing this type of supervised algorithm to be nominated as a help in the dermatopathological diagnosis of MM. In particular, analyzing the discrepancy between the performance of the FRF algorithm and the diagnosis by the dermatopathologist, a value of 17% was found, which is slightly lower than that described in the interobserver variability (about 25–26%). These data are similar to that reported by *Hekler* et al. [18] who, in their paper, trained a CNN according to a binary model of classification of nevi and melanoma, and reported a discrepancy value between the trained CNN and dermatopathologist equal to 18% for melanoma, 20% for nevi and 19% overall. On the other hand, it is important to underline that any AI algorithm is trained on diagnostic criteria chosen “a priori” by the pathologist and, therefore, there can similarly be false-positive misdiagnoses: this is the case with Spitz nevi. Indeed, *Hart S.N.* et al. [19] trained a CNN for binary classification between conventional nevi and Spitz nevi. Their algorithm was tested on WSIs of Spitz nevi and conventional nevi, producing a classification accuracy of 92% overall, based on a sensitivity of 85% and specificity of 99%. In a second phase of their study, the authors demonstrated how important it was to improve the diagnostic performance values of the AI algorithm to allow a dermatopathologist to preselect the areas of interest of the entire WSI, as in the absence of this, the validation accuracy was unacceptable at only 52%.

All these data agree in allowing us to consider the inclusion of the FRF algorithm in the normal workflow of histopathological diagnostics. The framework of the FRF algorithm used in this work is based on Weka [20] libraries implemented in the Java language.

The main motivation is in the novel approach adopted for the algorithm training. Other studies use a traditional approach based on the discrimination of the training model and of the testing model, which are performed by adopting different images [21]. In the proposed approach, it adopts the same image both for the training and for the testing: each image is converted into grayscale images, and in the same image [12] the classes training the model are identified (classes of similar features, including those of the Spitz nevi). In this way the doctor can circle the areas (classes) without having a historical dataset of images. The algorithm versatility gives an idea of the anomalous distribution (enhancing anomalous groups of pixels) of malignant areas quickly by estimating the covering percentage.

Furthermore, from a clinical point of view, there have already been numerous papers published in the literature that have attempted to correlate, specifically, dermoscopic characteristics with clinical diagnosis. A very interesting contribution by Argenziano et al. [22] in 2003 evaluated the sensitivity, specificity and, therefore, diagnostic accuracy of criteria commonly used for the diagnosis of MM. In the paper, 108 melanocytic lesions were evaluated by 40 experienced dermoscopists in an attempt to evaluate the interobserver and intraobserver agreement by using four algorithms such as the pattern analysis, ABCD rule, Menzies method and 7-point checklist. Of all these, there was a good agreement except for with the dermoscopic criteria, demonstrating how important it is also in dermoscopy to adopt more “accurate” parameters.

### Limitations

It is important to note that in our paper we used exclusively histopathological (morphological) criteria and, therefore, were subject to limitations including the different sizes of cancer cells, a concept intrinsic to tumor heterogeneity [23].

## 5. Conclusions

The FRF images were processed by following a specific image diagnostic protocol, oriented on reading and algorithm error minimization. An important tool for melanoma diagnosis is the probability image estimated by the processed FRF output image. The probability image is useful to better discriminate information about ambiguous lesions. A single probability image refers to a particular class of “defect” and enhances, by the white color, the defect distribution in the whole analyzed image. By knowing the dimension of the acquired microscope image, it is also possible to estimate the defect distribution percentage. All the adopted approaches are suitable to create a specific image vision platform for telemedicine digital pathology.

## Figures and Tables

**Figure 1 diagnostics-12-01972-f001:**
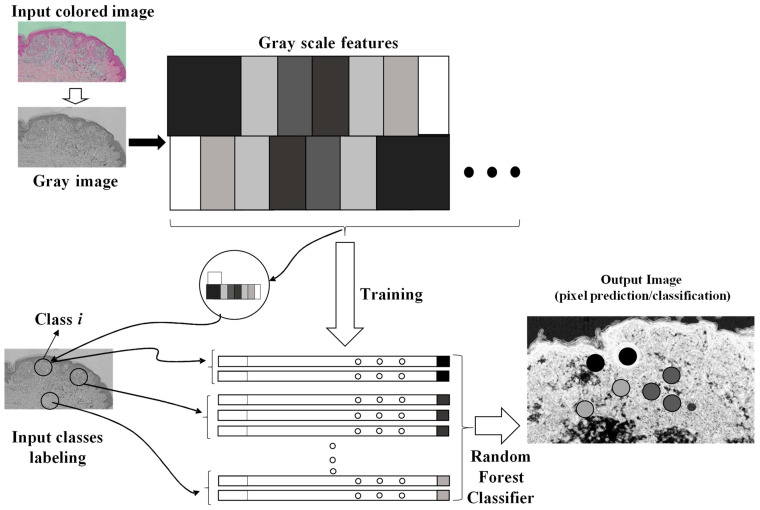
Image processing procedure based on pixel feature training and random forest classifier.

**Figure 2 diagnostics-12-01972-f002:**
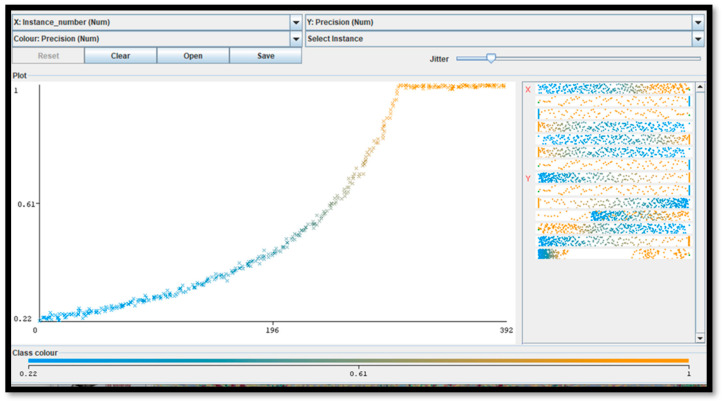
This figure illustrates the error metric (precision parameter [5]) of the adopted FRF algorithm versus iteration number (instance number) by proving that the final results are characterized by the maximum precision.

**Figure 3 diagnostics-12-01972-f003:**
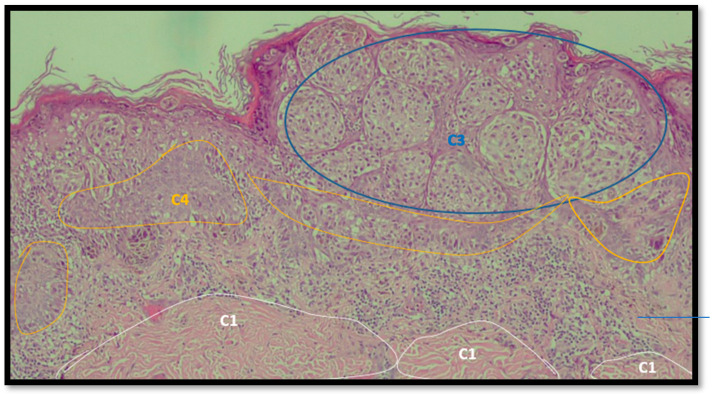
Example of analysis and selection of defects (such as architectural and cytological atypia, pagetoid spreading, possible ulceration) in the macro-area of an image of malignant melanoma. Note the different colors of the circles/ellipses used to subclassify the anomalies (defects).

**Table 1 diagnostics-12-01972-t001:** Summary of these criteria.

Dysplastic Nevus	Malignant Melanoma
*Architectural criteria (mandatory, major)*	
Lentiginous or contiguous melanocytic hyperplasia	Poor circumscription of the intraepidermal melanocytic component of the lesion
Focal melanocytic atypia	Increased number of melanocytes, solitary and in nests, within and above the epidermal basal cell layer and within adnexal epithelia (pagetoid spreading)
	Marked variation in size and shape of the melanocytic nests
	Absence of maturation of melanocytes with descent into the dermis
	Melanocytes in mitosis
*Architectural criteria (minor, at least 2)*	
“Shoulder phenomenon”	Melanocytes with nuclear atypia
Fusion of epithelial cones	Necrosis or degeneration of melanocytes
Subepidermal concentric lamellar fibrosis	

**Table 2 diagnostics-12-01972-t002:** Example of analysis of two micrographs of malignant melanoma in which some constituent elements of the Allen–Spitz criteria have been analyzed, such as: symmetrical or asymmetrical lesion, clustering of melanocytes in nests or presence of single melanocyte, and eventual pagetoid spreading.

Original Image	Defect Type (Name)	Defect Cluster (Enhanced Probability Image)	Percentage Presence on the Whole Image	Extension [mm^2^]
IMG00131 EE	//	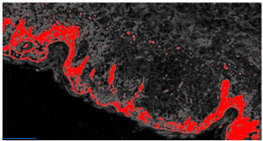	5.3%	0.106
IMG00132 EE	//	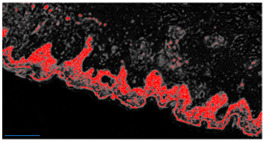	4.1%	0.082

**Table 3 diagnostics-12-01972-t003:** Example of two other images whose cytological characteristics were studied, including: cellular atypia, eventual pagetoid spreading, mitosis and nuclear pleomorphism.

Original Image	Defect Type (Name)	Defect Cluster (Enhanced Probability Image)	Percentage Presence on the Whole Image	Extension [mm^2^]
IMG00150	//	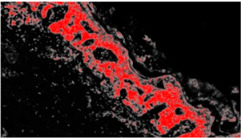	6.6%	0.132
IMG00151	//	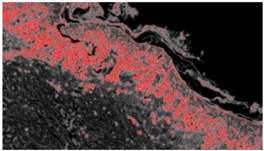	8.8%	0.176

## Data Availability

Not applicable.

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
