# Peer review of "Dermatopathology of Malignant Melanoma in the Era of Artificial Intelligence: A Single Institutional Experience"

_diagnostics, 2022, doi:10.3390/diagnostics12081972_

Round 1
Reviewer 1 Report
This is a really interesting study on AI employment in recognizing potential melanoma fragments of tissue which may lead to the development of malignancy. I have a few minor concerns.
1. Please describe the numeric definitions in more details. e.g. what are the h1, h2, h3 and h4 in the Hessian filtering and further (p. 4)?
2. Please refer to the dermoscopic attitude collected by Argenziano et al. (JAAD 2003, doi: 10.1067/mjd.2003.281), in particular to the concept of the "melanoma triad". I find some similarity to your observations, although your being gathered and exploited in more details...
3. Could yoy explicitly address the problem of cellular heterogeneity, in particular , the size heterogeneity in your attitude? See Klos & Plonka ABP 2020 DOI: 10.18388/abp.2020_5500
Author Response
Reviewer n’1: This is a really interesting study on AI employment in recognizing potential melanoma fragments of tissue which may lead to the development of malignancy. I have a few minor concerns.
- Please describe the numeric definitions in more details. e.g. what are the h1, h2, h3 and h4 in the Hessian filtering and further (p. 4)?
Answer n’1: Dear Reviewer n’1, first of all, thank you very much for your compliments: we are very happy and enthusiastic. Of course, we have specified the numerical definitions in more detail as per you asked. We hope it will go well.
Reviewer n’1: 2. Please refer to the dermoscopic attitude collected by Argenziano et al. (JAAD 2003, doi: 10.1067/mjd.2003.281), in particular to the concept of the "melanoma triad". I find some similarity to your observations, although your being gathered and exploited in more details...
Answer n’2: Dear Reviewer n’2, thank you very much. Our paper focused on the histopathology of malignant melanoma rather than on clinical-dermoscopic criteria for diagnosis of MM; nevertheless, we were pleased to read, studied, quoted and discussed the paper by Argenziano et al.
Reviewer n’1: 3. Could yoy explicitly address the problem of cellular heterogeneity, in particular , the size heterogeneity in your attitude? See Klos & Plonka ABP 2020 DOI: 10.18388/abp.2020_5500
Answer n’3: Dear Reviewer n’1, thousand thanks for this interesting and wonderful paper. We have read, studied, quoted and discussed the paper by creating a new small paragraph called "limitations" in which we address this issue. Thanks again for all your advice.
Reviewer 2 Report
Dear Authors,
Thank you for preparing this interesting manuscript. As the incidence of various skin cancers occurrence arises with each year and their diagnosis is still often based on a subjective assessment of the dermatopathologist, it is very important to develop new technologies to increase the efficiency with which these malignancies are recognized and subjected to treatment. This article perfectly fits this topic. I believe that such articles can bring a great value to the medical field and can open up a still poorly developed path for the implementation of machine learning techniques (artificial intelligence) focused on skin diseases into standard diagnostics.
In my opinion, the research part of this study is well-planned, but the manuscript requires corrections.
1. The first sentence of the introduction part (lines 49-54) is much too complicated to understand within a single read. Please divide it into shorter sentences.
2. Considering sentence (lines 62-66) ‘Instead, as was predictable, the difficulty in diagnosing ambiguous lesions, such as dysplastic nevi and malignant melanoma, together with the lack of interobserver agreement among dermatopathologists, has led to an objective difficulty in training artificial intelligence algorithms as well as those based on Machine Learning (ML) to a totally reliable, reportable and repeatable level [3,4].’ I wonder why is it so? What causes that melanoma is so hard to train those algorithms? A deeper description would be required in this part of the introduction.
3. The introduction part is generally to short. As some defects (important for the study) are introduced in the data acquisition part, I would recommend prolonging the introduction by a more detailed description of some features.
4. Taking into account that the Authors write in lines 83-84 ‘the study was approved by local ethical committee’, the two points 'Institutional Review Board Statement' and 'Informed Consent Statement' (line 309-310) were incorrectly described as 'Not applicable' whereas actually they should be completed with the note including the full name of the Local Bioethics Commission and number of the received consent, as well as an information that from all patients a written consent was obtained for this study. Please correct this.
5. Line 106, why this subsection is called 'material and methods: FRF....' when the whole section is focused on materials and methods? Please correct this.
6. There is a mistake in numbering of the equations - number 6 is missing.
7. Please, unify the style of writing the points in lines 150-153.
8. I would suggest a small reorganization of Figure 1. The ‘input colored image’, as it is received first and then implemented in the procedure, should be placed above (or next to the left) and not under the gray image.
9. Actually, Figure 4 and 5 should be renamed as tables, as they are tables...
10. Microscopic images lack scale bars.
11. In figure/table no. 4 and 5, numbers should be written with a dot (not a comma).
12. Please unify the writing of words in quotation marks – in the abstract the Authors used ‘..’, in the main text in general “…”.
13. Line 244, ‘in situ’ should be written in italics.
14. In my opinion, the text fragment from the discussion (lines 231-249) would fit better with the introduction part.
15. The final result has only just been described in the discussion part (lines 261-263; ‘In particular, analyzing the discrepancy between the performance of FRF and the diagnosis by the dermatopathologist, a value of 17% was found, slightly lower than that described in the inter-observer variability (about 25-26%).’), where it should be present in the results part in a proper form (graph or table) and a description. At this point, I feel unsatisfied with the results presented.
16. I would recommend to include the information that 'the same images were used for both training and testing' (line 283-284, discussion) in the methodological part first, and then it can be mentioned again in the discussion part.
17. Finally, please double check the revised manuscript as here a few grammar and punctuation errors occurred.
With kind regards.
Author Response
Reviewer n’3: Dear Authors,
Thank you for preparing this interesting manuscript. As the incidence of various skin cancers occurrence arises with each year and their diagnosis is still often based on a subjective assessment of the dermatopathologist, it is very important to develop new technologies to increase the efficiency with which these malignancies are recognized and subjected to treatment. This article perfectly fits this topic. I believe that such articles can bring a great value to the medical field and can open up a still poorly developed path for the implementation of machine learning techniques (artificial intelligence) focused on skin diseases into standard diagnostics.
Answer n’1: Dearest Reviewer n’3, thank you very much for these wonderful world: yes, the our goals is this! Thanks again, the authors
Reviewer n’3: 1. The first sentence of the introduction part (lines 49-54) is much too complicated to understand within a single read. Please divide it into shorter sentences.
Answer n’2: Dear Reviewer n’3, ok. We modified the sentence by reducing it in length and then dividing it into two separate sentences. We hope it will go well.
Reviewer n’3: 2. Considering sentence (lines 62-66) ‘Instead, as was predictable, the difficulty in diagnosing ambiguous lesions, such as dysplastic nevi and malignant melanoma, together with the lack of interobserver agreement among dermatopathologists, has led to an objective difficulty in training artificial intelligence algorithms as well as those based on Machine Learning (ML) to a totally reliable, reportable and repeatable level [3,4].’ I wonder why is it so? What causes that melanoma is so hard to train those algorithms? A deeper description would be required in this part of the introduction.
Answer n’3: Dear Reviewer n’3, thank you for your question. The question has its roots in the so-called "gray zone" of dermatopathology, ie that area of ​​interest in which the criteria used to define a frankly malignant lesion from the frankly benign become nuanced and difficult to define for EVERYONE. This leads to a real loss, or in any case reduction, of the inter-observer agreement, with difficult cases of atypical pigmented lesions that have been defined over time as MELTUMP (for example), or SAMPUS, or lesions with an uncertain biological behavior. This is the basic reason why something more objective is necessary and desirable, but, considering that any AI algorithm is based on criteria that are (at least at the beginning) provided by those who train, it was inevitable that this difficulty would also have repercussions. at the level of this field. We have also provided this explanation in the paper, as suggested.
Reviewer n’3: 3. The introduction part is generally to short. As some defects (important for the study) are introduced in the data acquisition part, I would recommend prolonging the introduction by a more detailed description of some features.
Answer n’4: Dear Reviewer n’3, we followed your advice.
Reviewer n’3: 4. Taking into account that the Authors write in lines 83-84 ‘the study was approved by local ethical committee’, the two points 'Institutional Review Board Statement' and 'Informed Consent Statement' (line 309-310) were incorrectly described as 'Not applicable' whereas actually they should be completed with the note including the full name of the Local Bioethics Commission and number of the received consent, as well as an information that from all patients a written consent was obtained for this study. Please correct this.
Answer n’5: Dear Reviewer, thank you very much. Our work was carried out in the context of a real Telemedicine center such as the CITEL of the Telemedicine Operations Center of the University of Bari "Aldo Moro". We added the information he suggested. Thanks again.
Reviewer n’3: Line 106, why this subsection is called 'material and methods: FRF....' when the whole section is focused on materials and methods? Please correct this.
Answer n’6: Dear Reviewer, sorry for the mistake. We correct. Always thanks.
Reviewer n’3: There is a mistake in numbering of the equations - number 6 is missing.
Answer n’7: Dear Reviewer n’3, it’s true. Sorry for mistake. Thanks again.
Reviewer n’3: Please, unify the style of writing the points in lines 150-153.
Answer n’8: Done.
Reviewer n’3: I would suggest a small reorganization of Figure 1. The ‘input colored image’, as it is received first and then implemented in the procedure, should be placed above (or next to the left) and not under the gray image.
Answer n’9: Done, thank you dear reviewer n’3.
Reviewer n’3: Actually, Figure 4 and 5 should be renamed as tables, as they are tables...
Answer n’10: Done, thousand thanks.
Reviewer n’3: 10. Microscopic images lack scale bars.
Answer n’11: Done, thanks again.
Reviewer n’3: 11. In figure/table no. 4 and 5, numbers should be written with a dot (not a comma).
Answer n’12: Done. Thank you.
Reviewer n’3: 12. Please unify the writing of words in quotation marks – in the abstract the Authors used ‘..’, in the main text in general “…”.
Answer n’13: Done, thank you.
Reviewer n’3: Line 244, ‘in situ’ should be written in italics.
Answer n’14: Done.
Reviewer n’3: In my opinion, the text fragment from the discussion (lines 231-249) would fit better with the introduction part.
Answer n’15: Dear Reviewer, we fully agree with you but, after the changes you suggested, we have already expanded the introduction by taking up some of the concepts discussed in this part of Discussion: so we thought we would leave it as it was. We hope you agree, otherwise we will change it.
Reviewer n’3: 15. The final result has only just been described in the discussion part (lines 261-263; ‘In particular, analyzing the discrepancy between the performance of FRF and the diagnosis by the dermatopathologist, a value of 17% was found, slightly lower than that described in the inter-observer variability (about 25-26%).’), where it should be present in the results part in a proper form (graph or table) and a description. At this point, I feel unsatisfied with the results presented.
Answer n’16: Dear Reviewer n’3, we consider your observation to be very fair. Therefore, we have taken steps to add this important information to the end of the results paragraph. We hope it will go well.
Reviewer n’3: 16. I would recommend to include the information that 'the same images were used for both training and testing' (line 283-284, discussion) in the methodological part first, and then it can be mentioned again in the discussion part.
Answer n’17: Done.
Reviewer n’3: Finally, please double check the revised manuscript as here a few grammar and punctuation errors occurred.
Answer n’18: Done, thanks again for all dear reviewer n’3.
Reviewer 3 Report
This is an interesting and well written manuscript with conclusion consistent with the experimental data. However the diagnosis of melanoma is based on immunohistochemical and molecular criteria and only on morphological criteria. Consequently I do not believethat this approach could be nominated in the histopathological diagnostic flow-chart of malignant melanoma. This should be clearly explained in the manuscript.
Author Response
Dear Reviewer n'2, thank you very much for your compliments and also thanks for your clarifications. We changed the conclusions slightly by adding this important detail. Thank you againRound 2
Reviewer 2 Report
Dear Authors,
thank you for improving your manuscript and for incorporating most of my recommendations.
Small clarification: while adding 'scale bars' not only should a line be included, but also a value with a unit.
With kind regards.
Reviewer 3 Report
None
Author Response
Thank you very much.
The authors